# Molecular Characterization of *Staphylococcus aureus* Isolated from Raw Milk Samples of Dairy Cows in Manhiça District, Southern Mozambique

**DOI:** 10.3390/microorganisms9081684

**Published:** 2021-08-08

**Authors:** Nilsa Nhatsave, Marcelino Garrine, Augusto Messa, Arsénia J. Massinga, Anélsio Cossa, Raúl Vaz, Angelina Ombi, Tomás F. Zimba, Hélder Alfredo, Inácio Mandomando, Cesaltina Tchamo

**Affiliations:** 1Instituto Superior de Ciências de Saúde (ISCISA), Maputo 977, Mozambique; nilsanhatsave1@gmail.com (N.N.); angelinaombi@gmail.com (A.O.); 2Centro de Investigação em Saúde de Manhiça (CISM), Maputo CP1929, Mozambique; marcelino.garrine@manhica.net (M.G.); augusto.junior@manhica.net (A.M.J.); arsenia.massinga@manhica.net (A.J.M.); anelsio.cossa@manhica.net (A.C.); 3Global Health and Tropical Medicine (GHTM), Instituto de Higiene e Medicina Tropical (IHMT), Universidade Nova de Lisboa (UNL), 1349-008 Lisbon, Portugal; 4Microbiology Laboratory, Hospital Central de Maputo (HCM), Maputo 1653, Mozambique; raulbrandle@gmail.com (R.V.); zimbatf@gmail.com (T.F.Z.); 5Faculdade de Ciências de Saúde, Universidade Zambeze, Tete 1018, Mozambique; helderjr80@gmail.com; 6Instituto Nacional de Saúde (INS), Ministério da Saúde, Maputo 1120, Mozambique; 7Department of Clinics, Veterinary Faculty, Universidade Eduardo Mondlane, Maputo CP257, Mozambique

**Keywords:** dairy cows, raw milk sample, *S. aureus*, antibiotic resistance, molecular typing, PVL, enterotoxins, Manhiça, Mozambique

## Abstract

Staphylococcal infections are among the most common foodborne diseases. We performed the antibiotic susceptibility and molecular characterization of *S. aureus* from milk samples of dairy cows in Manhiça District. We observed a high frequency of *S. aureus* (41%, 58/143), in which 71% (41/58) were from commercial farms and 29% (17/58) from smallholder farms. Half of the isolates (50%, 29/58) were resistant to at least one antibiotic, with higher rates of resistance to penicillin (43%, 25/58), followed by tetracycline (16%, 9/58). Multidrug-resistant and methicillin-resistant *S. aureus* isolates were rare (5%, 3/58 and 3%, 2/58, respectively). The genetic diversity was low, with predominance of human-adapted strains being: ST1/CC1-t5388 (78%) and ST152-t1299 (10%), followed by ST8/CC8-t1476 (5%) and ST5/CC5-t002 (3%) and lastly, ST508/CC45-t331 and ST152-t355, with 2% each. The Panton–Valentine leukocidin (*PVL*) gene was detected among 14% (8/58) of the isolates, while genes encoding staphylococcal enterotoxins were scarce (3%, 2/58). Our findings revealed a high frequency of *S. aureus*, with high rates of resistance to the antibiotics commonly used in veterinary and human medicine. Further investigations focusing on the molecular epidemiology of *S. aureus* from cattle and farmers will provide detailed insights on the genetic relatedness between the strains.

## 1. Introduction

Staphylococcal foodborne disease (SFD) is one of the most common foodborne diseases and a major concern in public health programs worldwide [1]. SFD results from the consumption of contaminated foods by *Staphylococcus aureus* enterotoxins (SEs) that are resistant to heat treatment [2,3]. The foods that have been frequently implicated in SFD are meat and meat products, poultry and egg products and milk and dairy products [4].

In addition to causing SFD in humans, *S. aureus* is an opportunistic pathogen responsible for causing conditions that range from superficial skin infections to life-threatening diseases [5]. On the other hand, in animals, it is the common agent of mastitis, causing significant financial losses to dairy farms [6]. Specific strains of *S. aureus* can possess different virulent traits predisposing to different clinical outcomes in the host, persistence of infections and antibiotic treatment response [7]. The treatment of infections caused by *S. aureus* has been challenged by the emergence of multidrug-resistant (MDR) strains, including methicillin-resistant *S. aureus* (MRSA) [8], as the result of indiscriminate use of antibiotics both in human and veterinary medicine [9,10]. Many of the genes encoding antibiotic resistance reside on mobile genetic elements, which can be exchanged between strains from animals and humans occupying the same ecological niche [11].

Several molecular typing methods (multilocus sequence typing (MLST), *spa* typing, *SCCmec* typing and pulse-field gel electrophoresis (PFGE)) are widely used to track epidemiologically related strains, allowing us to trace the origin of SFD [12]. These typing methods have revealed evidences of humans and cattle sharing the same strains of *S. aureus*. In addition, of greater concern is the emergence of supposed bovine-adapted strains in the human population as well as of supposed human-adapted strains in the bovine population, including the circulation of MDR strains [13].

While antibiotic susceptibility and molecular characterization of *S. aureus* isolated from the food chain have been extensively studied in developed countries, the corresponding data from Africa is limited. Available data have shown *S. aureus* isolated from milk samples with high rates of resistance to antibiotics used in veterinary practices among African isolates [14,15], as well as the detection of indistinguishable *S. aureus* isolated from cattle and humans, suggesting bacterial transmission between hosts [16].

Previous studies on the antibiotic susceptibility and molecular characterization of *S. aureus* in Mozambique only focused on patients with community-acquired infections, nosocomial infections or in carriers [17,18,19,20,21,22], while data of *S. aureus* isolated from the food chain were not investigated. Therefore, in this study, we determined the antibiotic susceptibility and the clonal structure of *S. aureus* isolated from milk samples of dairy cows in the Manhiça District, Mozambique between April and May 2019.

## 2. Materials and Methods

### 2.1. Study Design

The study was conducted in the Manhiça District, a rural area located about 80 km north from downtown Maputo (southern Mozambique). Manhiça has a tropical climate with two seasons, a warm, rainy season between October and March, and a cool and dry season during the rest of the year [23].

Between April and May 2019, we performed a cross-sectional study in one commercial farm and among six smallholder farms in the Manhiça District during the milking season. Because of the limited number of commercial farms in the district and limited accessibility to smallholder farms, sample size calculation could not be determined.

The commercial farm and the smallholder farms were attributed an identification code for anonymization. The commercial farm included in this study commercializes milk and dairy products (cheese, yogurt) in the Manhiça District. The milk from the smallholder farms is either commercialized directly to the customers (local residents/regional travelers) or to commercial farms. The milking process in the commercial farm is either mechanized or by hand, while in the smallholder farms, only hand milking is performed.

### 2.2. Specimen Collection

We performed physical examination of the udder quarters of each animal for viability, and only those in the lactation phase and with at least one viable teat were included in the study; therefore, the milk sampling per animal ranged from one to four teats. Animals in the colostrum phase or in the final phase of lactation were excluded. None of the cows from this study presented any clinical symptoms related to mastitis or were under antibiotic therapy.

The udder quarters with viability were cleaned with cotton soaked in water and soap, ending by disinfection of each teat with 70% alcohol. For each teat, the first three milk jets were discarded, and an aseptic milk sample (approximately 4 mL) was collected in a sterile tube (one tube per teat), conditioned at 2–8 °C and transported to the microbiology laboratory of the Maputo Central Hospital for culture and identification.

### 2.3. Isolation and Identification of Staphylococcus aureus

For *S. aureus* isolation, each sample was homogenized by vortexing, and a sterile 10 µL plastic loop was introduced on each tube containing milk and then cultured into blood agar plates and incubated at 37 °C in a 5% CO_2_ atmosphere for 24 h. All positive cultures with a Gram stain compatible to staphylococci were subcultured into blood agar plates and incubated overnight as mentioned above. Distinct isolates with morphology resembling staphylococci were selected from each blood agar plate for catalase testing. Catalase positive isolates were verified by Pastorex Staph-Plus testing (Bio-Rad, Marnes-la-Coquette, France) and confirmed with the BD BBL™ Coagulase Plasma, Rabbit “tube coagulase” test (BD, Sparks, MD, USA). All the isolates identified as *S. aureus* by conventional microbiology were stored in Microbank™ medium and sent to the microbiology laboratory of the Centro de Investigação em Saúde de Manhiça (CISM) for molecular confirmation and downstream analysis.

### 2.4. Preparation of Bacterial DNA and Molecular Confirmation of S. aureus

At CISM, the isolates were retrieved from the storage and cultured on blood agar plates and incubated overnight at 37 °C in a 5% CO_2_ atmosphere. Afterwards, one colony was selected from the blood agar plate and inoculated into 5 mL of BD Tryptic Soy Broth followed by overnight incubation at 37 °C. Upon the incubation, 1 mL of the culture was centrifuged, and the pellets were suspended and lysed in Tris-EDTA buffer with Triton-X 100 (Aldrich Chemical Co., Milwaukee, WI, USA) and proteinase K (Sigma Chemical Co., St Louis, MO, USA) for 1 h at 56° C, followed by boiling at 100 °C for 10 min as previously described [24]. The crude DNA was diluted in distilled water (1:10) and analyzed by conventional polymerase chain reaction (PCR) for confirmation of *S. aureus* species through the detection of the specific *nuc* gene which encodes the thermostable nuclease using primers and conditions previously described [25]. The reference strains *S. aureus* strains ATCC^®^ 25923^TM^, and *Staphylococcus epidermidis* ATCC^®^ 12228^TM^ were included as positive and negative controls, respectively. The DNA of confirmed *S. aureus* isolates was stored at −20 °C for further molecular characterization.

### 2.5. Antimicrobial Susceptibility Test

The molecularly confirmed *S. aureus* isolates were tested for antimicrobial susceptibility using the Kirby–Bauer disk diffusion and E-test methods, and interpreted according to the Clinical Laboratory Standards Institute (CLSI) guidelines (30th Edition) [26]. The isolates were tested against antibiotics used in veterinary medicine as well as antibiotics relevant to human medicine. The cefoxitin disk was used as a surrogate for oxacillin, for the prediction of putative methicillin-resistant *S. aureus* (MRSA phenotype). Inducible clindamycin resistance (ICR) was tested for all isolates resistant to erythromycin and susceptible or intermediate to clindamycin. Multidrug-resistant (MDR) phenotype was defined as resistance to ≥3 unrelated classes of antibiotics [27]. *S. aureus* strains ATCC^®^ (25923 ^TM^ and 29213 ^TM^) were used as quality controls.

### 2.6. Screening of Resistance Determinants

*S. aureus* isolates showing non-susceptibility profiles were screened for the corresponding resistance determinants (*tetM*, *tetL*, *ermA*, *ermC*, *msrA*, *dfrG* and *dfrA*(*S1*) by conventional monoplex PCR using specific primers and thermal cycling conditions (Appendix A). PCR for the main resistance determinants encoding for resistance to tetracycline (*tetK*), cefoxitin (*mecA*) and penicillin (*blaZ*) was performed among all the isolates regardless of the observed phenotype (Appendix A). The amplification products were separated through a 1.5% agarose gel stained with ethidium bromide. The 1-kb plusA 100-bp DNA ladder (Bio-Rad) was used as a molecular size marker in the gels. The *S. aureus* strain ATCC^®^ 25923 ^TM^ was used as a negative control.

### 2.7. Detection of Virulence Genes

The *S. aureus* isolates were analyzed by conventional PCR using primers and conditions previously established for detection of staphylococcal enterotoxins (*sea* to *see*), toxic shock syndrome toxin (*tst*) [28] and Panton–Valentine leukocidin (PVL) [29] related genes. The *S. aureus* 111066.2 was used as a negative control [18].

### 2.8. Staphylococcal Protein A (spa) Typing

We performed molecular typing by amplifying and sequencing the hypervariable region of the *spa* gene as described elsewhere [30]. The *spa* types were then assigned using the Ridom Staph Type version 2.2.5 (Ridom GmbH, Würzburg, Germany).

### 2.9. Multilocus Sequence Typing (MLST)

MLST was performed for all the isolates, using the scheme previously described [31]. The allelic profiles, sequence types (STs) and clonal complexes were assigned using the MLST *S. aureus* database (https://pubmlst.org/; accessed on: 20 May 2021).

### 2.10. Data Analysis

The statistical analyses were performed using STATA version 14.2 (StataCorp LP, College Station, TX, USA). Categorical variables were compared using the Chi-square or Fisher’s exact test when appropriate. We deemed *p*-value of 0.05 or lower to be statistically significant. The antimicrobial susceptibility data were entered in a spreadsheet and analyzed through WHONET version 19.8.6 (World Health Organization, Geneva, Switzerland).

The Simpson’s index of diversity (diversity index (DI)) was used to evaluate the genetic diversity [32].

## 3. Results

### 3.1. Frequency of S. aureus Isolation

One hundred and forty-three milk samples were collected from forty cows, in which twenty-eight (70%) cows were confirmed positive to *S. aureus*, with a similar distribution between cows from the commercial and smallholder farms (71%, 20/28 vs. 67%, 8/12, *p* = 1.000). The overall frequency of positive samples for *S. aureus* was 40% (57/143), with two *S. aureus* with distinct phenotypes isolated from one sample (Figure 1). The number of positive samples ranged among the 28 cows: (i) 7 positive samples were collected from 1 teat among 7 cows; (ii) 28 samples from 2 teats among 14 cows; (iii) 18 samples from 3 teats among 6 cows; lastly, (iv) 4 samples from 4 teats in 1 cow (Appendix A).

### 3.2. Antibiotic Susceptibility

Half of the *S. aureus* isolates (50%, 29/58) were resistant to at least one antibiotic class. The MDR phenotype was rare, with three isolates (5%), showing resistance simultaneously to: (i) PEN-TCY-ERY-CD; (ii) PEN-ERY-CD; and (iii) PEN-SXT-TCY. The highest frequencies of resistance were observed for penicillin (43%), followed by tetracycline (16%), and the lowest for erythromycin/clindamycin and co-trimoxazole with 3% and 2%, respectively (Table 1). The two distinct isolates (based on colony morphology) from the same sample showed distinct resistance patterns (resistance for penicillin vs. tetracycline).

Resistances to penicillin and to tetracycline were significantly more frequent in isolates from the smallholder farms than in the commercial farm (Table 1). All the isolates were susceptible to cefoxitin (MRSA phenotype not observed), ciprofloxacin, chloramphenicol, gentamicin, nitrofurantoin, daptomycin, linezolid and vancomycin.

### 3.3. Resistance Determinants

#### 3.3.1. Resistance Determinants on Resistant Isolates

We found consistent correlation between the phenotypic resistance and its resistance determinants. Approximately half of the isolates resistant to penicillin (48%, 12/25) carried the *blaZ* gene encoding for beta-lactamase. Among the isolates resistant to tetracycline, all (*n* = 9) carried the *tetK gene*, while 67% (6/9) and 11% (1/9) carried the *tetL* and *tetM* genes, respectively. One of the two *S. aureus* isolates with the ICR phenotype simultaneously carried the *mrsA* and *ermC* genes, while the other one carried only the *ermC* gene; the *ermA* gene was not detected. The unique isolate resistant to co-trimoxazole carried the *dfrG* gene, while the *dfrA(S1)* gene was absent.

#### 3.3.2. Resistance Determinants on Susceptible Isolates

All the isolates susceptible to penicillin (*n* = 33) were negative to the *blaZ* gene. On the other hand, 3/49 (6%) of the isolates susceptible to tetracycline were positive to the *tetK* gene. Two of the fifty-eight (3%) isolates phenotypically susceptible to cefoxitin harbored the *mecA* gene (MRSA strains).

### 3.4. Molecular Typing

The molecular typing revealed low diversity (DI = 0.39), in which isolates from the same *spa* type belonged to the same ST. We identified six *spa* types and five STs among the 58 *S. aureus* isolates: the strain ST1/CC1-t5388 (76%, *n* = 44) predominated, followed by ST152-t1299 (12%, *n* = 7), ST8/CC8-t1476 (5%, *n* = 3), ST5/CC5-t002 (3%, *n* = 2) and lastly, the ST508/CC45-t331 and ST152-t355 with 2% (*n* = 1) each.

Some strains were found exclusively among the smallholder farms (ST8/CC8-t1476, ST152-t355 and ST508/CC45-t331) or in the commercial farm (ST5/CC5-t002), as well as scattered in both (ST152-t1299 and ST1/CC1-t5388). *S. aureus* isolated from distinct teats of the same animal frequently belonged to the same clone (*n* = 16 animals) rather than in distinct ones (*n* = 5 animals), as shown in Appendix A.

The two isolates from the same sample which presented distinct phenotypes (colonies morphology) belonged to distinct strains (ST1/CC1-t5388 vs. ST152-t1299). The two isolates genotypically identified as MRSA belonged to the clones ST1/CC1-t5388 and ST508/CC45-t331, and were pan-susceptible and penicillin-resistant, respectively. We did not find an association between genotypes and resistance patterns (Table 2).

### 3.5. PVL and Staphylococcal Enterotoxins (SEs) Gene Detection

The frequency of the gene encoding for PVL toxin was low (14%, 8/58), and detected exclusively among the *S. aureus* isolates belonging to the ST152. The overall frequency of strains positive to SEs was low (3%, 2/58), in which one isolate (ST5/CC5-t002) was positive for the *sea* gene, while another (ST508/CC45-t331) for the *see*. None of the isolates were positive for the *tst* gene.

## 4. Discussion

This is the first study on molecular characterization of *S. aureus* isolated from raw milk of dairy cows in Mozambique. Our findings revealed high rates of *S. aureus* resistant to the antibiotics commonly used in human and veterinary medicine in Manhiça District, as well as the circulation of similar genotypes causing childhood bacteremia in the study community.

The high frequency of *S. aureus* with similar proportions between cows from a commercial farm and the smallholder farms suggests endemicity of *S. aureus* among farms in Manhiça District. Comparable prevalences of *S. aureus* from dairy cows have been reported in other regions of Africa [33,34], Europe [35,36,37] and Asia [38,39]. The isolates showed high rates of resistance to the antibiotics commonly used in human and veterinary medicine (e.g., penicillin and tetracycline) in Manhiça District, a setting with limited access to the second line antibiotics. The high rates of resistance to penicillin and tetracycline observed among *S. aureus* isolated from children admitted with bacteremia in the same setting [18,40] corroborate our findings suggesting the selective pressure of resistant *S. aureus* due to overuse of these antibiotics in both medicines, which is not surprising. The phenotypic resistance to antibiotics other than penicillin and tetracycline was uncommon or absent among the isolates from our study, possibly due to limited availability of those antibiotics in veterinary medicine [41].

The higher resistance to penicillin and tetracycline observed among the smallholder farms compared to the commercial farm in our study could be related to the sector of production, in which the smallholder farms in Manhiça District have less access to veterinary assistance associated with the indiscriminate use of antibiotics (by farmers or other personnel who are not certified veterinarians), as those can be obtained without prescription. On the other hand, on the commercial farm veterinary assistance is frequent, although some degree of antibiotics overuse is observed. Similar to our study, high rates of resistance to penicillin and tetracycline were previously reported among *S. aureus* isolated from milk samples of dairy cows in Ethiopia (84–86% and 27–54%, respectively) [14,15] and in Brazil (83% and 71%, respectively) [42]. On the other hand, a recent study conducted in South Africa which analyzed *S. aureus* from bovine milk samples and from nasal swabs of farmers in close contact reported higher rates of resistance to penicillin (75% vs. 29%) and tetracycline (42% vs. 3%) in humans than in cattle [41].

We detected two isolates of *S. aureus mecA*-positive (ST508/CC45-t331 and ST1/CC1-t5388) which were cefoxitin susceptible, and three isolates *tetK*-positive which were tetracycline susceptible, possibly due to non-expression of the related genes, resulting in underestimation of the burden of resistant strains in the food chain [43]. The MRSA strains from our study were distinct from the well-known zoonotic livestock-associated MRSA (LA-MRSA) strains belonging to the clonal complex (CC) 398 [41]. However, the rare detection of MRSA strains from our study is in accordance with previous reports on *S. aureus* isolated among patients from Manhiça District and from other regions in Mozambique (1–9%) [18,19,20,40], suggesting homogeneous epidemiology of MRSA strains in humans and animals in Mozambique. Similarly, the low circulation of MDR and the absence of MRSA were reported in studies conducted in Ethiopia [14] and South Africa [16,41]. In our study, half of the isolates resistant to penicillin (52%, 13/25) were negative to detection of the *blaZ* gene, possibly due to a mutation of the primer-annealing site that prevented amplification [41]. The detection of other resistance determinants correlated well with the phenotypic resistance exhibited, suggesting that those are the main encoding for the antibiotic resistance observed in our context.

Data on molecular typing revealed that some strains of *S. aureus* can be herd-specific forming a close niche [44], be endemic or even found dispersed in a region [17]. Although there was low diversity of *S. aureus* observed in our study, some strains were found exclusively among the smallholder farms or in the commercial farm, while others circulated in both sectors.

We observed a high clonal structure among the isolates from our study, in which isolates from the same *spa* type belonged to the same ST. In addition, *S. aureus* isolated from distinct teats from the same animal frequently belonged to the same strain than from a distinct one, suggesting contagious transmission in the spread of the bacteria [45,46], primarily via the milking machine or milkers’ hands [13]. These sources may play an important role in cow-to-cow transmission of the same [44,47] or distinct [48] strains of *S.*
*aureus* from milk or teat skin. We found two distinct strains (ST1/CC1-t5388 vs. ST152-t1299) isolated from the same milk sample, suggesting that the teats of the same animal can be colonized by unique or multiple strains, predisposing humans to infections by multiple strains of *S. aureus* through the ingestion of contaminated milk. In the Manhiça District, the milk commercialized by the commercial farm is pasteurized, while the one from the smallholder farms is often sold without any treatment, increasing the risk of infection for consumers. *S. aureus* can produce enterotoxins that are resistant to the milk treatment, predisposing humans to staphylococcal food poisoning [3] or clinical syndromes, such as toxic shock syndrome [5]. In our study, the frequency of SEs was low (3%), with only two isolates positive for *sea* and *see* genes, each. However, we only screened for genes coding for *sea* to *see*, and *tst*; further studies should include additional virulence factors (*sef* to *seo*, *seq*, *ser* and *seu*; hemolysins (e.g., *hla* and *hlb*); and exfoliative toxins (e.g., *eta* and *etb*)).

It is noteworthy that all the *spa* types but t355 and t5388, and all the STs from our study have previously been reported in staphylococcal bacteremia in children admitted in the Manhiça District Hospital [18,40], raising the hypothesis of potential anthroponotic transmission of *S. aureus*; however, additional studies are needed to verify this hypothesis. The Manhiça District is a semi-rural area where there is frequent contact between cattle and humans. Therefore, future studies in our setting should investigate the *S. aureus* strains isolated from cattle as well as from individuals working in close contact with these animals in order to understand the genetic relatedness of strains between hosts. In addition, our study had a limited sample size, so future studies should consider exploring additional Mozambican districts and provinces known as major dairy producers, as well as other foodborne disease-associated pathogens.

The strains from our study were previously isolated in milk samples of dairy cows in studies conducted in distinct regions across Africa (ST1, ST8, ST508 and t407) [14,16], Europe (ST1, ST5 and ST152–355) [49], Asia (ST1) [50] and in the Americas (ST1) [51]. All those STs have been isolated in dairy cows as well as in human colonization or infections [13], suggesting bacterial transmission between hosts.

In our study, the *PVL* gene (encoding for PVL cytotoxin) was found exclusively in the ST152 (t355 and t1299) strain, a strain that possibly originated from Africa, and then expanded throughout Europe [52]. The cytotoxin displays a specific host tropism [53], being frequently detected and associated with necrotic lesions involving the skin or mucosa in humans [29], suggesting human-to-cow transmission of *S. aureus* [16]. A study conducted in Ethiopia reported that all the strains belonging to t335 isolated from dairy cows were positive for the *PVL* gene [14].

## 5. Conclusions

Our findings revealed high frequency of isolation of *S. aureus* in raw dairy milk in the Manhiça District. The isolates showed high rates of resistance to the antibiotics commonly (penicillin and tetracycline) used in veterinary and human medicine. We observed low diversity among our isolates, in which some strains were found exclusively among the smallholder farms or in the commercial farm, as well as scattered in both. It is noteworthy that all the *spa* types but t355 and t5388, and all the STs from our study were reported as being associated in children admitted with bacteremia in the Manhiça District Hospital. Further investigations focusing on the molecular characterization of *S. aureus* isolated from cattle and farmers will provide detailed insights on the genetic relatedness between strains.

## Figures and Tables

**Figure 1 microorganisms-09-01684-f001:**
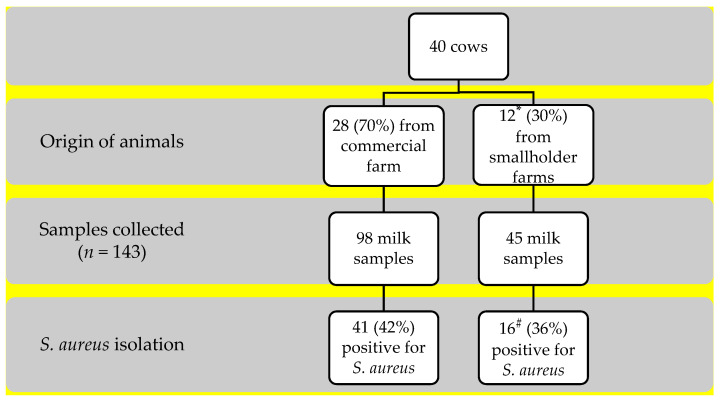
Samples collected and frequency of *Staphylococcus aureus* isolation. * Average of two cows/smallholder farm. # In one sample two *S. aureus* isolates with distinct phenotypes were found.

**Table 1 microorganisms-09-01684-t001:** Antibiotic resistance of *Staphylococcus aureus* isolated from raw milk samples of dairy cows in southern Mozambique between April and May 2019.

Antibiotics	Commercial Farm *n* = 41 (%)	Smallholder Farms *n* = 17 (%)	*p*	Total *n* = 58 (%)
Penicillin	12 (29)	13 (76)	0.001	25 (43)
Tetracycline	3 (7)	6 (35)	0.014	9 (16)
Erythromycin/Clindamycin ^1^	1 (2)	1 (6)	0.504	2 (3)
Co-trimoxazole	0	1 (6)	NA	1 (2)

^1^ The erythromycin-resistant isolates exhibited the inducible clindamycin resistance (ICR) phenotype. NA: Not applicable.

**Table 2 microorganisms-09-01684-t002:** Clonal relatedness and resistance profile of *S. aureus* isolated from raw milk in dairy cows in Manhiça District, southern Mozambique.

			N° Isolates Per Sector			
*Spa* type (*n*)	MLST (*n*)	*PVL* Gene (*n*)	Smallholder Farms (*n*)	Commercial Farm (*n*)	Resistance Pattern (*n*)	Resistance Determinants (*n*)
t5388 (44)	ST1/CC1 (44)	0	8	36	PEN (13); PEN-TET (1); TET (1)	*blaZ* (1); *tetK* (2); *blaZ*-*tetK* (1), *mecA*-*tetK* (1)
t1299 (7)	ST152 (7)	7	4	3	PEN (2); TET (3); PEN-TET (1), PEN-TET-ERY-CD, (1)	*blaZ* (1); *blaZ*-*ermC*-*tetK*-*tetL* (1); *blaZ*-*tetK*-*tetL* (1); *tetK*-*tetL* (2); *tetK*-*tetM*-*tetL* (1)
t1476 (3)	ST8/CC8 (3)	0	3	0	PEN (1); PEN-TET (1); PEN-TET-SXT (1)	*blaZ* (1); *blaZ*-*tetK*-*tetL* (1); *blaZ*, *tetK*, *dfrG* (1)
t002 (2)	ST5/CC5 (2)	0	0	2	PEN (2)	*blaZ* (2)
t331 (1)	ST508/CC45 (1)	0	1	0	PEN	*blaZ*-*mecA*-*tetK*
t355 (1)	ST152 (1)	1	1	0	PEN-ERY-CD	*blaZ*-*msrA*-*ermC*

PEN: Penicillin; TET: Tetracycline; ERY: Erythromycin; CD: Clindamycin; SXT: Co-trimoxazole.

## Data Availability

All relevant data are within the paper and its Appendix A.

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
