# Peer review of "Molecular Characterization of Staphylococcus aureus Isolated from Raw Milk Samples of Dairy Cows in Manhiça District, Southern Mozambique"

_microorganisms, 2021, doi:10.3390/microorganisms9081684_

Round 1
Reviewer 1 Report
- Lines 104-106: How much of the milk was collected, 4ml? Please clarify it here.
- Lines 110-111: Did the samples were centrifuged to concentrate the bacteria before using for the isolation? The sample volume is really matter with the isolation, for the concentration of S. aureus should be very low if large volume of milk sample was used here.
- Line 122: Please described the protocol used here for DNA extraction and confirmation of S. aureus briefly with one or two sentences, although more details can be found in the cited references.
Was the negative control was used in the DNA extraction and PCR assays? Please clarify what kind of negative control was used here, and other assays throughout the article.
- Since lots of primers were used in this study, the authors should list all of them in tables like table S1 which will be much helpful for the readers.
- Lines 257-261: Please italicize the bacteria and gene name, and check throughout the manuscript.
- Is there some clinical symptoms or other diseases progressed on these animals who were positive for S. aureus? How about the age of these animals if the related information is available?
- Did the other sample types, such as animals-related samples or milk products from these farms studied here, were collected to confirm the existence of S. aureus?
If not, it is misleading to use the title with “anthroponotic transmission”, because there was no data directly supported this. The detection of S. aureus, which was also found in the human community, in the milk from the dairy farm is not sufficient to draw the conclusion.
Reviewer 2 Report
Abstract informs me that the manuscript requires English editing which should be completed/required before resubmission. This is a relatively small study on the epidemiology of S. aureus isolated from diary cattle, the fact that it is completed in Africa is of interest to many.
Major
- Site description can be placed understudy design section.
- Line 89-90. The authors need to explain the inability to measure sample size? Seems like you can state the number of farm operations tested and how many strains were isolated? esp considering lines 187-188? Fig 1? Table 2?
- Were strains isolated from cows with obvious mastitis or any cows under antibiotic therapy? Would be good to know if health care workers are present in the operations that find MDR strains? Or perhaps since they are not MRSA, this is not required?
- Also what antibiotics are used on the farms that were analyzed? Which ones are approved for use in Africa?
- Line 234 - some were cefoxitin susceptible but harbored mecA gene? Genome examination? Could exposure of these strains to methicillin lead to the expression of methicillin resistance?
- Discussion should be greatly reduced for conciseness
- Line 325. How often are multiple unique isolates acquired from the same teat or cow?
- Line 337. Transmission - I think more proof is required to demonstrate interspecies transmission - rather similar strains are found in children and dairy cattle....
Minor.
- Line 110. Emulsified???? Sentence is unclear.
- Lines 137-142. Disks and concentration descriptions not required, results allow the reader to recognize what drugs were tested.
- Lines 276-277. Overuse? How can the authors make this claim? It might just be use? In humans is penicillin and tetracycline sold without prescription? are the same drugs used as growth promotants or used to treat mastitis?
Round 2
Reviewer 1 Report
All the concerns were addressed in the revision.
Reviewer 2 Report
The authors should once again reconsider more English corrections before final submission.